# The Advances in Glioblastoma On-a-Chip for Therapy Approaches

**DOI:** 10.3390/cancers14040869

**Published:** 2022-02-09

**Authors:** Arielly H. Alves, Mariana P. Nucci, Javier B. Mamani, Nicole M. E. Valle, Eduarda F. Ribeiro, Gabriel N. A. Rego, Fernando A. Oliveira, Matheus H. Theinel, Ricardo S. Santos, Lionel F. Gamarra

**Affiliations:** 1Hospital Israelita Albert Einstein, São Paulo 05652-000, Brazil; ariellydahora1997@gmail.com (A.H.A.); mariana.nucci@hc.fm.usp.br (M.P.N.); javierbm@einstein.br (J.B.M.); nicolemev@gmail.com (N.M.E.V.); eduardafribeiro@usp.br (E.F.R.); gabriel.nery@einstein.br (G.N.A.R.); fernando.anselmo@einstein.br (F.A.O.); nutrimatheusht@gmail.com (M.H.T.); ricardosds@gmail.com (R.S.S.); 2LIM44-Hospital das Clínicas da Faculdade Medicina da Universidade de São Paulo, São Paulo 05403-000, Brazil

**Keywords:** glioblastoma on-a-chip, glioblastoma model, microfluidic devices, tumor cells co-culture, therapy glioma

## Abstract

**Simple Summary:**

This systematic review showed different therapeutic approaches to glioblastoma on-a-chip with varying levels of complexity, answering, from the simplest question to the most sophisticated questions, in a biological system integrated in an efficient way. With advances in manufacturing protocols, soft lithography in PDMS material was the most used in the studies, applying different strategy geometrics in device construction. The microenvironment showed the relevant elaborations in co-culture between mainly human tumor cells and support cells involved in the collagen type I matrix; remaining an adequate way to assess the therapeutic approach. The most complex devices showed efficient intersection between different systems, allowing in vitro studies with major human genetic similarity, reproducibility, and low cost, on a highly customizable platform.

**Abstract:**

This systematic review aimed to verify the use of microfluidic devices in the process of implementing and evaluating the effectiveness of therapeutic approaches in glioblastoma on-a-chip, providing a broad view of advances to date in the use of this technology and their perspectives. We searched studies with the variations of the keywords “Glioblastoma”, “microfluidic devices”, “organ-on-a-chip” and “therapy” of the last ten years in PubMed and Scopus databases. Of 446 articles identified, only 22 articles were selected for analysis according to the inclusion and exclusion criteria. The microfluidic devices were mainly produced by soft lithography technology, using the PDMS material (72%). In the microenvironment, the main extracellular matrix used was collagen type I. Most studies used U87-MG glioblastoma cells from humans and 31.8% were co-cultivated with HUVEC, hCMEC/D3, and astrocytes. Chemotherapy was the majority of therapeutic approaches, assessing mainly the cellular viability and proliferation. Furthermore, some alternative therapies were reported in a few studies (22.6%). This study identified a diversity of glioblastoma on-a-chip to assess therapeutic approaches, often using intermediate levels of complexity. The most advanced level implemented the intersection between different biological systems (liver–brain or intestine–liver–brain), BBB model, allowing in vitro studies with greater human genetic similarity, reproducibility, and low cost, in a highly customizable platform.

## 1. Introduction

Glioblastoma (GBM) is the most common primary malignant brain tumor in adults. The annual incidence of GBM in the United States is 3.23 cases per 100,000 people [1] and is one of the most fatal malignant diseases in humans. The patient median survival is around 14–16 months, and the relative survival is only five percent in five years. Tumor resection surgery, radiotherapy, and chemotherapy are the usual treatments for GBM [2,3], however due to their invasibility, heterogeneity, and inefficacy in medication carriage over the hematoencephalic barrier (BBB), this disease continues to exhibit failure in the face of conventional treatments [4].

Alternative therapy approaches have been proposed to overcome the GBM treatment difficulties, such as immunological therapy, using vaccine compounds of peptides, dendritic cells, adoptive T cells, or chemical immunological checkpoint inhibitors (PD-1, CTLA-4), as well as virotherapy to regulate the immune tumor response [5]. Gene therapies, on the other hand, aim to alter the genetic structure of target cells, resulting in improved immune responses, reprogramming of the tumor microenvironment (TME), and angiogenesis normalization [6]. The use of nanoparticles in nanobiotechnology has primarily aimed to overcome the challenges of delivering genes and drugs to the tumor location. Furthermore, when combined with an alternating magnetic field, nanoparticles’ physical and magnetic features have been used to regulate metabolic processes and produce hyperthermia [7]. Additionally, photodynamic treatment (PDT), which causes molecular instability by heat stress, has shown that biochemical modulation can occur through the generation of reactive oxygen species (ROS) [8]. As highlighted in this study, numerous therapeutic techniques have been used alone or in combination to try to improve GBM therapy responses.

Large amounts of investments are spent examining the effectiveness of therapeutic agents for treating tumors in the search for novel therapies or improvements to existing treatments for tumors, although in many cases, more than 90% effectiveness is not obtained in clinical research [9,10]. One of the important reasons for this problem is the use of platfors that do not satisfactorily predict many of the proposed clinical treatments [11] as there are important limitations in these platforms: in vitro 2D and 3D, in animal models, and in silico.

Many studies have used conventional cell culture (2D), spheroids, and 3D co-culture to test different therapeutic approaches. These models are commonly used to carry out in vitro studies because they are simple to implement, low cost, high yield, and have a low ethical problem, but cell efficiency may decrease due to inappropriate interactions between cells and cells-extracellular matrix (ECM) [12], lack of vascularity, no predictive power, and no shear stress, which can cause changes in cell phenotype during in vitro culture [11,13,14]. In vivo models, on the other hand, have served as a preclinical preview of the translational pattern. Animal tumors, primarily in rodents, are the primary tool for elucidating biochemical and physiological processes in living organisms prior to clinical testing in humans [15,16]. These models allow the assessment of cell migration, invasion, growth, proliferation, angiogenesis, immune responses, drug toxicity, and the effectiveness of multiple therapeutic approaches. In addition, this platform shows more physiological, genetic similarity, moderate prediction of drug behavior, physiological microenvironment, enable mutation studies, and with some limitations, such as being highly expensive, requiring specialized personnel and facilities, low-throughput; no prediction for humans, inability to mimic human-specific features, long-term culture, ethical issues, among others [17]. Despite its capabilities, medications evaluated in preclinical cancer trials had a success probability of only 3.4% through phase I clinical trials, with a failure rate of 54% in the final stages, owing to insufficient effectiveness and poor safety [18].

Organ-on-a-chips have developed as a new testing option that offers a promising way to overcome the limitations of traditional in vitro and preclinical models. There are considerable disparities between existing models and humans in the replication of genetic, metabolic, physiological, and pathological complexity, according to evidence. The organ-on-a-chips platform, according to studies, can more accurately predict the efficacy and reactions to drugs and therapeutic processes than in vitro and in vivo testing [19]. In addition, some cell types cultured in 2D may be more sensitive to the toxic effects of drugs than when cultured in an organ-on-a-chip architecture [20]. Thus, the organ-on-a-chips recreate cell–cell or cell–ECM interactions, spatiotemporal physicochemical gradients, or the dynamic properties of the cellular microenvironment [21,22,23]. As a result, the complex organ-on-a-chips design enables in vivo replication of microenvironments, offering a stable platform for nanomedicine evaluation [24], presenting itself as a low cost alternative, preservation cell phenotype, customized design, control on physical and biochemical properties of the tumor microenvironment, well-defined vessel endothelium, gradient compatible, dynamic system, control of hydrodynamic parameters, real-time measurement, and microcopy compatible [11].

Organ-on-a-chips applied to cancer research [25,26,27,28,29,30,31,32] address models which involve several aspects, such as tumor growth, angiogenesis, cell invasion, intravasation, extravasation, and metastasis [33,34,35], as well as physiological drug exposures [36], which can assess disease progression and contribute to the development of precision medicine and personalized treatments using tissue from patients [37].

Given these aspects, this systematic review aimed to verify the use of microfluidic devices in the process of implementing and evaluating the effectiveness of therapeutic approaches in GBM tumors in the PubMed and Scopus literature databases. The fabrication of microfluidic devices, the composition of the microenvironment and the ECM, tumor cells, and support cells in culture were evaluated; as well as therapeutic approaches used, and the techniques applied to assess the effectiveness, providing a broad view of advances to date in the use of this technology and their perspectives.

## 2. Methods

### 2.1. Search Strategy

This systematic review followed the Preferred Reporting Items for Systematic Reviews and Meta-Analyses (PRISMA) guidelines [38]. The publication search was performed between February 2011 and February 2021, indexed in the PubMed and Scopus databases using the following Boolean operators (DecS/MeSH), and keywords sequence for each database:

PubMed: (((((Glioblastoma[Title/Abstract]) OR GBM[Title/Abstract])) OR Glioma[Title/Abstract])) AND (((((((((“organ-on-a-chip”[Title/Abstract]) OR “human-on-a-chip”[Title/Abstract]) OR microfluidic[Title/Abstract]) OR “organs-*on-chips*”[Title/Abstract]) OR “organs-on-a-chip”[Title/Abstract]) OR “microfluidic device”[Title/Abstract]) OR “lab-*on-chips*”[Title/Abstract]) OR “glioblastoma-on-a-chip”[Title/Abstract]) OR “GBM-on-a-chip”[Title/Abstract]);

Scopus: ((TITLE-ABS-KEY (glioblastoma) OR TITLE-ABS-KEY (gbm) OR TITLE-ABS-KEY (glioma))) AND ((TITLE-ABS-KEY (organ-on-a-chip) OR TITLE-ABS-KEY (human-on-a-chip) OR TITLE-ABS-KEY (microfluidic) OR TITLE-ABS-KEY (organs-*on-chip*) OR TITLE-ABS-KEY (organs-on-a-chip) OR TITLE-ABS-KEY (microfluidic AND device) OR TITLE-ABS-KEY (lab-*on-chip*) OR TITLE-ABS-KEY (glioblastoma-on-a-chip) OR TITLE-ABS-KEY (gbm-on-a-chip) OR TITLE-ABS-KEY (microvasculature-on-a-chip) OR TITLE-ABS-KEY (microenvironment AND in AND a AND chip) OR TITLE-ABS-KEY (brain AND cancer AND chip))).

### 2.2. Inclusion Criteria

We included only original articles published in English in the last ten years, with the available full text, and that used microfluidic devices to evaluate different therapeutic approaches for glioblastoma tumor models developed from the culture of tumor cells. 

### 2.3. Exclusion Criteria

We excluded articles that did not report the therapeutic approach in the microfluidic device, that did not perform a tumor microenvironment reconstitution from the use of tumor cells, as well as articles indexed in more than one database (duplicates), review articles, letters, articles in press, communications, book chapters, abstracts, incomplete articles, editorials, and expert opinions.

### 2.4. Data Extraction

In this systematic review, the collected data were segregated into the following topics: (i) the microfluidic device design, their material used, and its manufacturing method; (ii) the characteristics of the cells used in 3D culture and the medium; (iii) the microenvironment reconstitution for the glioblastoma model and their maintenance; and (iv) the therapeutic approaches applied in the devices and the techniques used for the therapeutic efficacy evaluation.

### 2.5. Data Analysis

The percentage distribution, obtained for each variable analyzed in the tables was used to characterize and present all of the results. Each study was classified into 3 categories of complexity, from (+) to (+++), based on how each topic was approached separately in each table. Finally, we considered the analysis of the results reported in the tables and applied a generic classification of device complexities in four categories (I–IV).

## 3. Results

### 3.1. Overview of the Reviewed Literature

We searched publications of the last 10 years, considering the period between September 2011 and September 2021, indexed in PubMed and Scopus, and a total of 446 articles were identified. Of the 119 articles found in PubMed, 94 were excluded after screening (89 duplicated in Scopus search, and 5 reviews), and 22 articles were excluded after eligibility analysis (12 articles did not report the therapeutic approach used for glioblastoma on-a-chip, 6 articles reported only the usage of the microfluidic device for analysis of the part of the experiment, such as CHIP-Seq or CHIP-qPCR, and 4 articles developed the study in silico), thus, only 3 articles were included from this database. Of the 327 articles identified in Scopus, after screening, 48 articles were excluded (18 reviews, 15 conference papers, 8 book chapter/series, 2 notes, 2 publications in other languages, 1 conference review, 1 editorial, and 1 short survey), and 260 articles were excluded after eligibility analysis (120 articles did not report the therapeutic approach used for glioblastoma on-a-chip, 110 articles reported only the usage of the microfluidic device for analysis of part of the experiment, such as CHIP-Seq, CHIP-qPCR, and chromatin immunoprecipitation—CHIP, and 30 articles developed the study in silico), thus, only 19 articles were included from this database. As shown in Figure 1, only 22 unduplicated full-text articles were included in this review [39,40,41,42,43,44,45,46,47,48,49,50,51,52,53,54,55,56,57,58,59,60], and the histogram and spider chart show the distribution of articles by year and research centers, respectively.

### 3.2. Design and Fabrication of Microfluidic Devices

Regarding the microfluidic device fabrication and its geometric characteristics, we analyzed the different materials used (organic or inorganic polymers) as well as the fabrication technology applied. These aspects reflect the complexity of devices used to improve the glioblastoma model and their therapeutic analysis, as shown in Table 1. Of the selected studies used in this study, 91% produced in-house devices [39,40,41,42,44,45,46,48,49,50,51,52,53,54,55,56,57,58,59,60], only 9% of studies used commercial devices [43,47], due to the variability of device design used in the research. Regarding technology used in this fabrication, different lithographic techniques were used, namely, 68.2% soft lithography [39,40,42,43,46,48,49,50,52,53,54,55,58,59,60], 13.6% photolithography [51,56,57], and 4.5% two-photon lithography [41], 4.5% used 3D-printing systems [44], and 9.1% did not report the technology used [45,47]. The main material of the devices was 88% polymers (72% PDMS [39,40,42,43,45,46,48,49,50,52,53,54,55,56,57,58,59,60], 8% polycarbonate [49,56], and 4% of PEGDA [51], and modified polyethersulfone [54]). Only 4% used inorganic synthetic polymer composed of silicon [44], and one study did not report the material used [47]. The most substrate used in the device production was glass in 64% of the selected studies [41,42,43,44,45,48,50,51,52,53,55,58,59,60], followed by 18% PDMS [40,46,49,54], and 5% no cover [39], and 14% did not report the substrate [47,56,57]. In relation to the mold material used in the device fabrication process, 68% of studies reported the use of SU-8 photoresist [39,40,41,42,43,46,48,49,50,52,53,54,56,57,58,59,60] due to the lithography technology that is commonly used this mold and in 4% of studies ethyl lactate was used [39] or SPR950/SF6 nanowires [57] or silicon [46]. Furthermore, 12% of studies did not report [45,47,55] and 8% did not do this process [44,51].

When the geometric characteristics of microfluidic devices was investigated, the selected studies reported mainly the fabrication of rectangular shapes (40.9%) for the culture region [42,47,48,49,50,52,55,56,57,58], following by 18.2% square [46,53,59,60] or circle shapes [40,43,44,51], and 4.5% reported the use of oval [54], semicircle [39], perimetric cylindrical pillars [41], or rectangular with circle array [48]. Furthermore, 4.5% did not report the shape of the culture region in the device [45]. The device design (dimensions and structure) varied a lot between studies, as well as the materials and methods of fabrication due to these parameters; we elaborate one analysis considering the level of device complexity from (+) to (+++) levels. Almost half of the studies (40.9%) were classified as level (++) complexity [39,43,45,46,48,51,53,57,59], which was considered shape with connections, simple material, and method, followed by level (+) (36.4%), which involved simple shape for culture [40,42,47,49,50,52,55,58], and level (+++) (22.7%), showed the more sophisticated method of fabrication [41,44,54,56,60], with multi-interfaces connected as depicted in Figure 2.

### 3.3. Cells Used in 3D Culture in Microfluidic Devices

We analyzed the cell characteristics (type, origin, and source) and their environment (medium culture and supplements) for glioblastoma on-a-chip model development in microfluidic devices. Regarding the model development, tumor and support cells were used isolated (63.6%) [40,42,43,46,47,48,50,51,53,57,58,59,60] or in co-culture (36.4%) [39,41,44,45,49,52,54,55,56] inside of the device (Table 2). 

Among the glioblastoma tumor cells, the most reported was U87-MG (40%) [39,40,41,44,48,52,53,58], including the use of this cell with modifications (U87-MG-GFP [41] and U87-MG/KD/SC [42]), then U251-MG (16%) [42,46,54,56], including their modifications (U251-MG/KD/SC [42] and induced U251-MG [46]), 16% C6 [43,47,59,60] (included the TMZ and BCNU resistant C6 [47]), 12% of GBM primary [39,44,49,51,55,57], 8% of GSC [49,57], 4% T98G [50], and F98-GFP [55]. In addition, some studies used different tumor cells, such as HepG2 (5.6%) [39,54], MCF7 (2.6%) [53], and Caco-2 (2.6%) [54], when compared to all tumor cells, and endothelial cells were used as microenvironment support, aiming at enriching the ECM, being 20% of HUVEC [44,52], hCMEC/D3 [41,56], and astrocyte cells [39,41], and 10% of each of the following cells: HBMEC [49], BMEC [39], Eahy926 [49], and Bend3 [55]. Of these cells, 80% were of human sources [39,40,41,42,44,45,46,48,49,50,51,52,53,54,56,57,58], being only 22.85% of primary culture [44,49,51,57], and 20% of animal sources [39,43,47,55,59,60] (22.22% of primary culture [39,55]). Only two studies that used the primary culture reported the number of passages that varied from 3 to 10.

Interestingly, the medium of culture used in microfluidic devices varied according to the type of culture, in co-culture, more than one type of medium culture was reported. The most used was DMEM (46.2%) [39,41,42,44,45,46,47,48,50,52,53,56,58,59,60], then DMEM-F12 [46,57], and RMPI-1640 with 15.4% [43,49,54,55,56]. These types of mediums were the same as those used in the 2D culture and added in the same proportion of cells seeded. In primary cell culture, the medium was supplemented with some growth factors (recombinant human EGF, FGF, LIF, EndoGRO-MV Supplement kit, astrocyte growth supplement, ECCS, FGFb, B-27, GA-1000, VEGF, hEGF, hFGF-β, R3-IGF-1) due to the complexity of culturing primary cells.

The complexity evaluation on cell culture—more than half of the studies were classified as level (++) due to using co-culture, spheroid, or modified cells culture [39,44,45,46,47,49,52,54,55,56], followed by level (+) classification (36.4%) with isolated culture use [40,42,43,48,50,51,53,57,58,60]. Only the study by Tricinci (4.5%) used co-culture associated with the spheroid, classified in level (+++) [41].

### 3.4. Methods Cultivation of Cells Used in the 3D Culture

Regarding the 3D culture of glioblastoma model in a microfluidic device (Table 3), the culture methodology that involves the ECM components (concentration and volume) and cell types (concentration, culture time, medium change, and their flow rate) that represent important aspects for the development of a tumor model biomimetic to evaluate the different methodologies and therapeutic agents, was analyzed.

Another relevant aspect in 3D co-culture is the ECM addition, which was reported in 73% of the selected studies [39,42,43,44,45,47,48,49,50,52,53,54,55,56,58,59]. Collagen type I represented 35%, being the most used in ECM composition [39,42,43,44,45,47,48,55,58,59], followed by 15% Matrigel [43,50,54,56], and a smaller proportion (4%) BdECM [44], fibronectin [49], TG-gelatin [52], GelMA [53], and agarose [56]. Of these studies, 45.8% reported ECM concentration used ranging from 0.1 to 12 mg/mL [39,42,43,44,45,48,50,53,54,55,56], and only 16% reported the volume administration, ranging from 8 to 20 µL [43,47,48,53]. In contrast, 20% did not use any ECM components [40,41,46,51,60]. The culture methodology in the microfluidic device focused mainly on the order and position of cell culture, from the treatment of the device with ECM to maintenance after the culture. Some strategies were used to promote the formation of a 3D matrix, inverting the device surface [45,48].

For glioblastoma model development, 51.3% of the selected studies used human cells (25.6% U87 [40,41,46,51,60], 10.3% U251 [42,46,54,56], 7.7% GSC [49,57], 5.1% human GBM [44,49,51,57], and 2.6% T98G [50]), followed by 12.8% of GBM from rats (10.3% C6 [43,47,59,60] and 2.6% F98-GFP [55]), and in 10.3% of studies different associated tumor types (carcinoma [39,54] and adenocarcinoma [54]) were used. Furthermore, 31.8% of studies used some supporting endothelial cells, with one or more of these cells combined (astrocytes, hCMEC/D3, HUVEC, HBMEC, BMEC, Eahy926, and Bend3), being prevalent in the co-culture with the first three of these cells (20% each), aiding in chemical communication and secretion of the ECM, obtaining results closer to those obtained in vivo experiments [61,62,63,64]. In relation to the cell type, we also analyzed the concentration used, which varied between the different types, as well as within the same type, for example, the U87 number cells ranged from 10^4^ to 10^7^ cells/mL. Cell culture time of 92% of the studies was from 0.125 to 10 days and medium change during culture was reported in 54% of the studies, being carried out from 2 to 72 h. The flow rate was reported in only 37.5% of the studies, ranging from 0.5 to 4.7 × 10^3^ µL/min, and 8.3% of studies did not apply the shear rate [56,59], an important factor for tumor growth.

The complexity evaluation of microenvironment construction showed that almost half (45.5%) of studies used simple ECM level (+) [40,42,43,46,47,51,57,58,59,60], followed by 27.3% level (++) [45,48,49,50,53,55] and (+++) [39,41,44,52,54,56], that match the use of two or more ECM combined, and ECM primary or synthetic scaffold, respectively.

### 3.5. The Efficiency of Glioblastoma Therapeutic Approach in the Microfluidic Device

Table 4 analyzed the different therapeutic approaches for glioblastoma through the microfluidic device and allowed it by microenvironment mimicking, combining the therapeutic, can increase the number of conditions to test, besides that the outcome observed its more similar to in vivo outcomes than in vitro experiments. Regarding the therapeutic approaches, the use of chemotherapy alone [39,40,41,42,44,45,46,47,48,49,50,51,52,54,55,56,57,59] or combined with other drugs [44,51,54] or conditions were reported in most of the selected studies (77.3%), followed by different combined therapeutic strategies (13.6%), such as phototherapy [55,59], and irradiation associated with drug delivery [51], as well as the therapeutic strategy in an isolated way, such as phototherapy (4.5%) and magneto hyperthermia therapy (4.5%) [43]. The drug most reported in the chemotherapy approach was an alkylating agent, the temozolomide (45.5%) [40,42,44,47,48,49,51,54,57] dose ranged from 0.005 to 1200 μM, following by 9.1% for doxorubicin (amphetamine) [50,55] that ranged from 0.03 to 1 μg/mL, and 4.5% of 24 different drugs with varied classes: chemical inhibitors (antibody-functionalized nutlin-loaded nanostructured lipid carriers, simvastatin, KU60019, methoxyamine, O^6^-benzylguanine, tamoxifen, irinotecan, and sunitinib), antioxidants (coenzyme Q10, resveratrol, catechins, α-lipoic acid, and ascorbic acid), alkylating agent (cisplatin and cyclophosphamide), antimicrotubular (paclitaxel and vincristine), antimetabolites (capecitabine and 5-fluorouracil), antibiotic (actinomycin D), antifungal (allicin), antibody (bevacizumab), microRNA (Anti-miR363), and siRNA (HIF1α/HIF2α inhibitor). The time of therapy varied from around 0.16 until 168 h.

The methods used to evaluate the therapeutic approaches include more than one technique in each study. Cell viability analysis was the most reported in the selected studies (95.45%), then 27.27% cell proliferation, 18.18% of oxidative stress, and migration/invasion of cells, and a further 4.54% for molecular characterization, DNA methylation, autophagy, metabolites, and permeability. Some techniques were used with more than one purpose of analysis, such as cell viability and proliferation using live/dead dye (25.71%) [39,41,42,43,49,53,54,56,57,59,60], immunostaining (25.71%) [41,42,46,48,54,55,58], CCK-8 kit (20%) [39,44,48,53,54], and CA/Pi (17.14%) [40,45,47,48,50]. The cell migration and invasion ware also available using 60% CA/PI dye [45,47,48], and 40% immunostaining (MMP2) [48,58]. The oxidative stress was analyzed by different techniques as CellROX Orange, DHE and NDA, SOSG, ROS, and GSH [47,52,54,59]. When analyzing the studies’ outcomes, between the chemotherapeutics utilized as well as the different therapeutic approaches and their impacts on the cellular microenvironment, the recommended therapies showed efficacy in various therapeutic approaches.

The complexity evaluation of therapeutic approaches in the glioblastoma on-a-chip model showed that almost half (45.5%) of studies used combinations of drugs (level ++) [39,40,42,47,48,51,52,54,57,58], following by 31.8% of level (+++) [41,43,44,53,55,59,60], that reported therapeutic approaches combined and less often (22.7%) the level (+), which used only one drug for chemotherapy [45,46,49,50,56].

We established a global classification of the glioblastoma on-a-chip model for therapeutic approaches, at different levels of complexity (I–IV), with level IV being the most complex, based on all aspects investigated in the present study and the results presented in the tables. The studies were classified considering their design and fabrication; cell culture isolated or co-cultures, ECM complexity, besides the therapeutic approaches used. This way, few studies (4.5%) were classified with low level of complexity due to their used simple shape, a single-cell type culture, without ECM, and a simple therapeutic approach [42,52]. Levels II and III often already had the most complexity reported with 36.4% [39,40,43,49,58] and 40.9%, respectively [45,47,48,50,51,53,54,55,57,59,60], shown to improve the design complexity through the use of the concentration gradient, as also parallel chambers with interconnections through pores, or the use of some type of ECM. Level III was regarded as the use of co-culture, advanced therapeutic approach, or the improvement of criteria used in level II. Of the studies, 18.2% were classified as level IV due to the use of intersection between different biological systems (liver–brain or intestine–liver–brain), BBB model, tri-culture, ECM adaptation, or use the synthetic scaffold [41,44,46,56].

In brief, Figure 3 shows the main aspects found in this systematic review, in which the development of microfluidic devices was more evident with the use of soft lithography technology (68.2%) [39,40,42,43,46,48,49,50,52,53,54,55,58,59,60] and the PDMS material (72%) [39,40,42,43,45,46,48,49,50,52,53,54,55,56,57,58,59,60]. Regarding the microenvironment, the main ECM used was collagen type I (35%) [39,42,43,44,45,47,48,55,58,59], followed by Matrigel (15%) [43,50,54,56], and 27% did not report the use of this type of scaffold. The tumor environment was made up mainly by U87-MG (40%) [39,40,41,44,48,52,53,58] from human glioblastoma cells and in 31.8% of co-culture, the use of support cells HUVEC [44,52], hCMEC/D3 [41,56], and astrocytes [39,41] with 20% each, was reported. The majority of therapeutic approaches evaluated the efficiency of some type of chemotherapy (77.4%) [39,40,41,42,44,45,46,47,48,49,50,51,52,54,55,56,57,59] through the cellular viability and proliferation, as also their migration, invasion, oxidative stress, autophagy, and permeability. Furthermore, some alternative therapies were reported in a few studies (22.6%) [43,44,53,55,59,60], even in conjunction with chemotherapy.

## 4. Discussion

In general, the glioblastoma on-a-chip models are developed based on the aims of the researchers, which reflect the diversity found in this systematic review from the manufacture of microfluidic devices to the reconstitution of the glioblastoma microenvironment in a 3D model, aiming at therapeutic approaches. Most of the microfluidic devices reported in the review were fabricated in house (91%) [39,40,41,42,44,45,46,48,49,50,51,52,53,54,55,56,57,58,59,60], being little used, the commercially available devices (9%) [43,47], due to the specificity of the aim to use, that requires a versatile design technology, capable of providing different combinations of microsystems for varied therapeutic approaches.

Regarding the device fabrication, the technology most applied for the development in the studies was lithography (86.4%) [39,40,42,43,46,48,49,50,51,52,53,54,55,56,57,58,59,60]; more evident being the use of soft lithography (68.2%) [39,40,41,42,43,46,48,49,50,52,53,54,55,58,59,60] and the material often used was polymers (88%) (PDMS, polycarbonate, PEGDA, and modified polyethersulfone), PDMS being the the most used (72%) [39,40,42,43,45,46,48,49,50,52,53,54,55,56,57,58,59,60]. The soft-lithographic technique is a simple, inexpensive, high throughput method for fabricating micrometer resolution patterns with good precision. However, for this procedure, another lithography method is necessary, such as photolithography or e-beam lithography, to fabricate the mold cast. The material most used for this mold was SU-8, a photoresist (68%) [39,40,41,42,43,46,48,49,50,52,53,54,56,57,58,59,60], that is popular for biological applications due to its high level of compatibility. However, for submicron resolution across two dimensions, photolithography or e-beam lithography is more adequate. Photolithography was the second most used technique in the review (13.6%) [51,56,57], being considered powerful not only to create a master mold but also as a stand-alone method that can offer micron-resolution patterns across a large area of the substrate [65].

In terms of device material, PDMS was the most used (72%) and it has a variety of advantages, including being durable, inert to most materials (patterned or molded), and chemically resistant to many solvents. However, this material also suffers from high compressibility, which causes a seal’s shallow relief features to deform, bend, or collapse. The molding step is facilitated by the elasticity and low surface energy of the PDMS, which also gives the possibility to replicate the size and shape of the features present in the mold by mechanical deformation. In addition, PDMS molds can be manufactured from a single master [66]. Among substrates, glass was most often in the reviews (64%) [41,42,43,44,45,48,50,51,52,53,55,58,59,60], followed by PDMS (18%) [40,46,49,54], nano/microfluidic glass channels giving improved control of the chemistry in the microsystem; PDMS or other polymers are already often used due to their low-cost fabrication process [67], but they are chemically active and strongly absorb proteins to their surface unlike glass channels, which are inert to most chemicals. Furthermore, glass channels are easy to clean, maintain, reuse, and very efficient in microscopic analysis due to optical characteristics [68].

For the glioblastoma model, the most commonly used cells include human-derived cell lines, such as U87 and U251, and mouse cell lines C6 and F98. U87 human GBM was the most reported in the selected studies (25.6%) [39,40,41,42,44,45,46,48,49,50,51,52,53,54,56,57,58] as an alternative preclinical testing model, following by the use of U251 [42,46,54,56] and C6 cells [43,47,59,60] (10.3% each). All these cell lines exhibit similar morphological characteristics regarding GBM nuclear pleomorphism and high mitotic index, except F98, which resemble anaplastic glioma. The most aggressive and invasive model is the F98, while the C6 has moderate invasiveness. U87 exhibits profuse neovascularization and has been used to study angiogenesis. When comparing U87 to U251, it was observed that the U87 cells exhibited a significantly higher rate in relation to their proliferation, invasion, and migration [69], and this difference was also observed in the 3D model, showing a rapid migration, and the highest invasion ability (the length of protrusions and the number of cells, invading into the collagen) [70]. For microenvironment studies, C6 has been well used because it resembles human GBM immune infiltrates, being considered a good model of an immunocompetent host for in vivo studies, due to its ability to cause a moderate immune response, as well as U87 and U251 cells. Other tumor cells, such as HepG2 cells, a liver hepatocellular carcinoma, were also used in the same device to compare the therapeutic efficacy and metabolization in different types of drugs and the interaction of the brain–liver system [39]. Another study also used these cells associated with Caco-2, human colorectal adenocarcinoma cells, to evaluate the intestine–liver–glioblastoma biomimetic system [54]. Already, MCF7 human breast cancer cells were used only in regard to glioblastoma therapeutic efficacy and cell migration [53].

The ECM is another relevant aspect for microenvironment formation inside the microfluidic device, since helps cells to attach, communicate, and provides physical scaffolding to biochemical and biomechanical processes, necessary for tissue morphogenesis, differentiation, homeostasis, and other cell functions [71,72]. Currently, a wide range from natural proteins to synthetic scaffolds has been used for culturing cells in a 3D environment. The choice of a suitable matrix depends on the cell type being used and specific experimental objectives. Materials of natural origin are commonly used, to mimic several key features of the native ECM, such as type I collagen, hyaluronic acid, laminin, fibronectin, gelatin, alginate, as well as Matrigel (ECM extracts) [73]. Another potential alternative to Matrigel is GelMA, a natural ECM reported in this review [53]. 

Of selected studies, 73% used some type of ECM, and collagen type I was the most reported (35%) [39,42,43,44,45,47,48,55,58,59], as well as being the most important ECM component with which cancer cells interact during their growth. It is the preferred substrate for the adhesion and migration of these cells and also stimulates their invasive behavior. This provides a strong rationale for the use of collagen I matrices in investigations pertaining to invasive behavior by cancers and metastasis [74]. The second component most used in the review was Matrigel (15%) [43,50,54,56], this material is chemically similar to the major components of basement membranes, imparting strength and integrity, but is unable to mimic the barrier function of intact basement membranes due to lesser resistance to cell penetration [74]. This material showed an influence on the spheroids’ formation due to its composition that contains a high percentage of laminin, collagen IV, enactin, proteoglycans, and growth factors, associated or not with other natural polymers (collagen, chitosan, hyaluronic acid) or synthesized (polyethylene glycol). Both materials can be used in different applications for generation of 3D models (organoids, primary tissue culture, embryonic stem cells, or induced pluripotent stem cells), tissue explants, and cellular differentiation, suggesting their importance in the composition and function of the matrix. Furthermore, the main components of the matrices cited above can influence the metabolism of cancer cells, as well as interfere with cell signaling and tumorigenesis [74,75]. 

In addition to ECM, some studies reported the use together with support cells as the astrocytes, HUVEC, and hCMEC/D3, for the formation of the 3D model in the microfluidic device. The supporting cells are able to modulate and produce ECM through secreted factors [76,77]. HUVEC was also used to evaluate the ability of angiogenesis [78,79]. Only the study by Tricinci [41] did not report the use of some type of ECM, substituted by the use of a synthetic scaffold as a similar function.

The HUVEC co-cultures with human glioma cells (U87-MG and T98) resulted in vascular sprout formation. However, no vascular sprout formation was observed in HUVEC co-cultured with human teratocarcinoma cells (NT2), which do not produce VEGF, suggesting their importance in the angiogenesis role. Hypoxia condition in gliomas is another way to lead to the upregulation of VEGF expression and angiogenesis, enhancing tumor growth through neovascularization [80]. EA.hy926 is another endothelial cell used in the neoangiogenesis model and has the advantage of being reproducible, not depending on the primary tissue nor having differences in response along with the genetic variation of each sample as HUVEC [81]. 

Some microfluidic devices in this review [39,40,56] are aimed at the blood–brain tumor barrier (BBTB) study in the microenvironment, since the molecular selectivity of BBB allow homeostasis in physiological conditions, as also shields the neoplastic cells by blocking the delivery of peripherally administered chemotherapies. For this BBTB, different types of support cells were used, such as hCMEC/D3, when in co-culture with astrocytes has been reported to restore some of the BBB-differentiated phenotypes of isolated brain endothelial cells (BECs) by having a particular impact on the expression and maintenance of tight junction (TJ) proteins [82]. However, the systematic review of human BBTB models of brain permeability for novel therapeutics [82] showed a great variability of cell origin (stem cell-derived, primary or immortalized), monoculture versus co-culture, and other parameters that affected model success. These aspects may cause the under or overestimation of drug permeability and therapeutic efficacy. Nevertheless, also highlighted by some important analyses, as the co-cultures of ECs with astrocytes, and pericytes had significantly upregulated protein or mRNA expression of tight and adherent proteins, and transporters, irrespective of whether pluripotent stem cell, primary, or immortalized cells lines were induced. The murine brain microvascular endothelial cells (bEND3), also used in this review [55], are known to be successful in forming barriers when co-cultured with astrocytes [83]. Astrocytes, in turn, have widely distinct morphological, molecular, and functional properties, suggesting the existence of heterogeneous subpopulations and have been identified as modulators of the BBB permeability, as well as their impact on TEER, and gene expression [84].

We also analyzed the methodological steps used in the formation of microenvironments within the device, being relevant to note that 77.3% reported details of the construction of this microenvironment, in which 31.8% of studies firstly covered the device with some type of ECM followed by cell culture, in a staggered isolated or in co-culture manner. At the same frequency, the simultaneous infusion of matrix and cells was observed. Only two studies (9.1%) reported the inversion of the PDMS surface to form the 3D structure [45,48], and the study by Qu et al. [45] reported the use of type I collagen after the formation of spheroids. In parallel, the flow of the medium responsible for the shear rate was described in only eight studies (36.3%), which is a relevant aspect regarding the cell phenotype, as well as for the renewal of the medium and nutrients.

Regarding the therapy approaches applied in the glioblastoma on-a-chip model, chemotherapy represented 77.3% in this review, through the main use of alkylating agent temozolomide (45.5%), alone [40,42,44,47,48,49,51,54,57] or in combination with other drugs [44,51,54]. Furthermore, the combined drugs aimed at the same action mechanisms, as the alkylating agent (CP), or other actions as antimetabolites (CAP, 5-FU, and actinomycin-D), antimicrotubular (PTX), antifungal (allicin), HMG-CoA reductase inhibitors (simvastatin), antioxidants (CoQ10 and Res), antibody (BEV), topoisomerase I inhibitor (CPT-11), and microRNA (Anti-miR363), that influence tumor development in different ways. Therefore, the current gold standard therapy for glioblastoma includes maximal safe surgical resection, followed by TMZ-based chemoradiotherapy [85]. A similar approach was adopted in the study by Yi et al. [44] that used a concurrent chemoradiation and temozolomide associated with other drugs.

New combined therapeutic approaches that involved using advanced technology were applied in this review, such as 9.1% of studies that used nanotechnology resources and focused ultrasound to improve the drug delivery in chemotherapy [41,55]. In addition, alternative approaches were used as photodynamic therapy (13.6%) [53,59,60], and 4.5% of them use gold and iron nanoparticles associated with near-infrared laser [53], and alternating magnetic field [43], respectively, that promote the death of tumor cells by hyperthermia. Despite the low frequency of studies with these alternative therapies, scientific interest in this area has grown due to technological advances and the development of multifunctional probes capable of being applied in translational studies, combining more than one therapy and analysis technology. The 3D model allows the application of these technological advances in models that use mainly human tumor cells, capable of predicting more mimetic responses than in vitro and in vivo studies.

Therefore, considering all aspects involved in this review, we used the classification system in four levels of glioblastoma on-a-chip fabrication complexity. The increased complexity in the elaboration of the devices verified in this review (18.2% level IV [41,44,46,56], 40.9% level III [45,47,48,50,51,53,54,55,57,59,60], and 36.4% level II [39,40,43,49,58]) has reflected the diversity of components present in the real tumor microenvironment, as well as the responses obtained from the interaction of the different systems, cells and ECM, as described in the study by Jo et al. [50] that observed chemoresistance regarding the use of Matrigel in the treatment with DOX, among other aspects. In this sense, the literature has proven that the responses of organ-on-a-chip studies have been increasingly closer to in vivo studies than the results obtained in in vitro [86,87], and represents an excellent platform for validation of therapeutic processes for glioblastoma tumors.

This review showed the current aspects of glioblastoma research through the organ-on-a-chip device for therapeutic approaches, but the diversity features of device elaboration and approaches did not allow us to conclude which was the most effective therapeutic approach among the studies, being a relevant limitation of this study. 

## 5. Conclusions

This systematic review identified the diversity of glioblastoma on-a-chip to assess therapeutic approaches, with different levels of complexity. We found that soft lithography, a printing process with a high micrometric resolution, and PDMS, a biocompatible and chemically resistant substance, were found to be the most used in this review. The tumor microenvironment was mainly composed of ECM rich in collagen type I associated with human tumor cells, cultivated in a 3D framework. Chemotherapy remains a more studied approach, alone or in comparison with other therapeutic alternatives, in the search for more efficient ways of drug delivery, with fewer collateral effects, using nanocarriers associated with drug activation techniques and hyperthermia promotion for treatment of glioblastoma.

In terms of experiment complexity, a few researches have shown a low level of complexity by using a simple shape, unicellular culture, without ECM, and a straightforward treatment strategy. The adoption of a concentration gradient, parallel chambers with interconnections, or some sort of ECM were among the most often reported intermediate complexity (level II and III) features. The usage of co-culture, an advanced therapeutic strategy, or an enhancement of the criteria employed in level II, were all explored in the third level. Interestingly, the most advanced level implemented the intersection of different biological systems (liver–brain or intestine–liver–brain), the BBB model, tri-culture, ECM adaptation, or the use of synthetic scaffolds, allowing for the recognition of advances in this technology for organ-on-a-chip studies with greater human genetic similarity, reproducibility, and low cost on a highly customizable platform. Thus, finally, we can conclude that, taking into account the studies included in the review, the glioblastoma-on-a-chip platform is an excellent alternative for evaluating the therapeutic process of this type of tumor with high reproducibility.

## Figures and Tables

**Figure 1 cancers-14-00869-f001:**
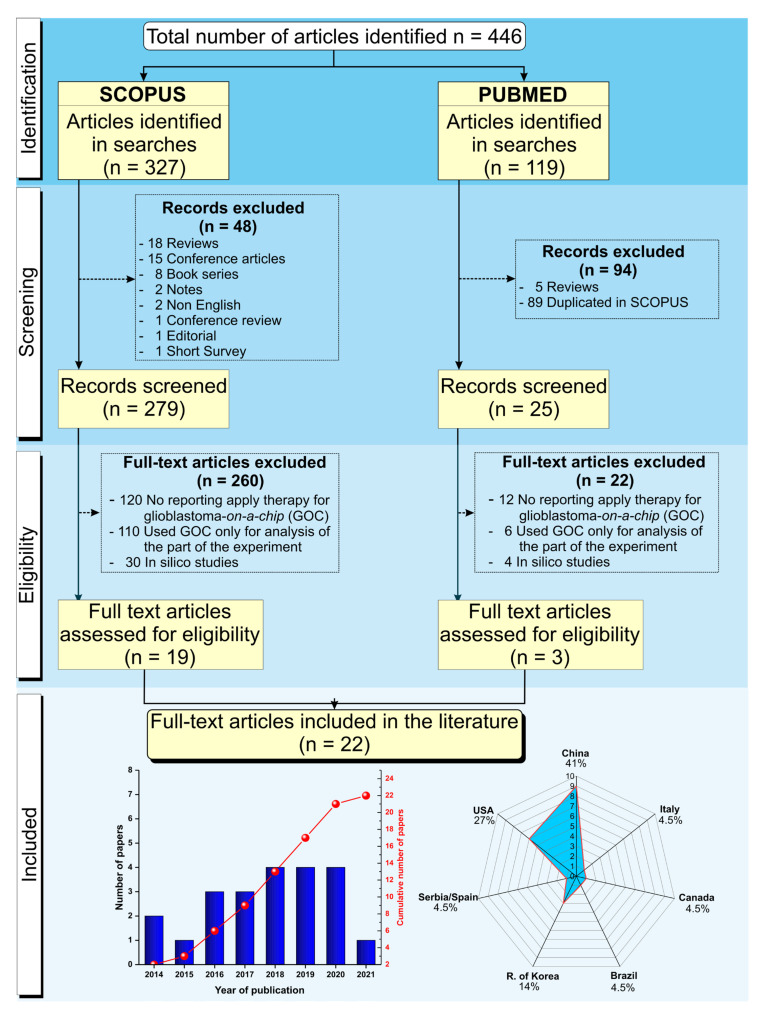
Schematic representation of the screening process of articles for inclusion in this systematic review following PRISMA guidelines from the identification of 446 studies in the SCOPUS and PubMed databases, following the predetermined inclusion and exclusion criterias. After initial screening and eligibility assessment, 424 were excluded, and only 22 studies were included in this review. The histogram contains the distribution of the 22 articles included by year of publication represented by blue bars and the representation of the cumulative growth until 2021 by the red points. The spider chart shows the regional distribution (countries) of the research centers, where the studies included in this review were developed.

**Figure 2 cancers-14-00869-f002:**
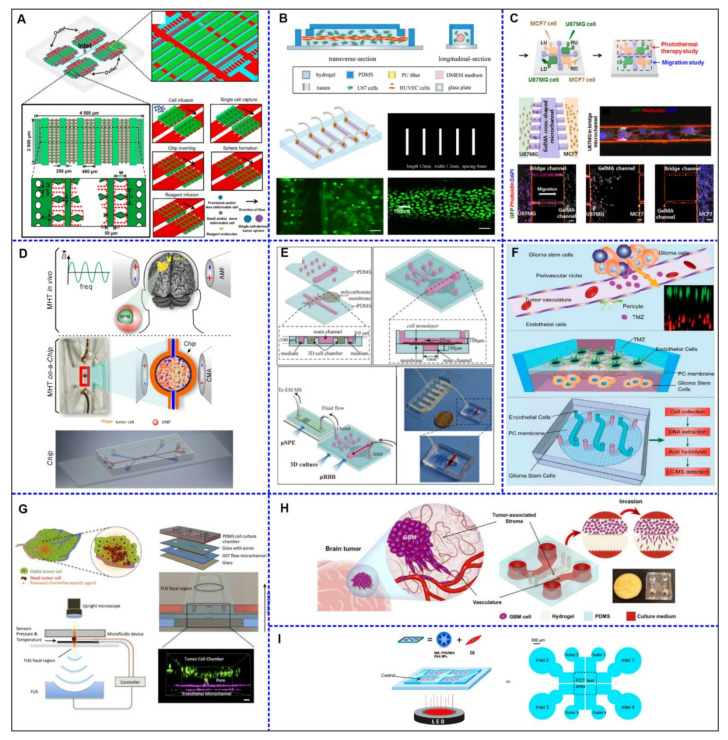
The schematic figures of glioblastoma on-a-chip devices for therapy approach used in some of the selected studies of this systematic review. (**A**) The integrated microfluidic system for single-cell separation and sphere formation, adapted with permission from [46], the American Chemical Society. (**B**) 3D co-culture unit generative process and the analysis of the confocal images of the chip, showing the HUVEC cells in the lumen, adapted with permission from [52], Analytica Chimica Acta. (**C**) MCF7 and U87MG cancer cells diagonally seeded into square-shaped microchambers, in the hydrogel microfluidic device, and analysis of confocal microscopy images, adapted with permission from [53], Electrophoresis. (**D**) Magnetohyperthermia process in tumor-on-a-chip using magnetic nanoparticles dispersed in aqueous medium submitted to an alternating magnetic field., adapted with permission from [5], Einstein. (**E**) A microfluidic platform mimics the blood-brain barrier (BBB) using two PDMS sheets a polycarbonate membrane. BBB unit was directly connected to the μSPE unit for mass spectrometry detection., adapted with permission from [56], Analytica Chimica Acta. (**F**) Biomimetic design of miniaturized artificial perivascular niche on a chip for analysis on chemoresistance in GSCs and endothelial cocultured and relative metabolites by liquid chromatography mass spectrometry, adapted with permission from [49], Analytical Chemistry. (**G**) The closed-loop acoustofluidic device with multilayer for drug release in a tumor by the focal ultrasound system, adapted with permission from [55], Small. (**H**) Glioblastoma on-a-chip comprised of tumor and tumor-associated stroma compartments with side channels (delivered nutrients and drugs), and the actual image of the fabricated model., adapted with permission from [42], International Journal of Molecular Sciences. (**I**) Simplified photodynamic therapy of methylene blue conjugated polyacrylamide nanoparticles, with a polyethylene glycol dimethacrylate cross-linker on microfluidic chip, adapted with permission from [59], Chemistry of Materials.

**Figure 3 cancers-14-00869-f003:**
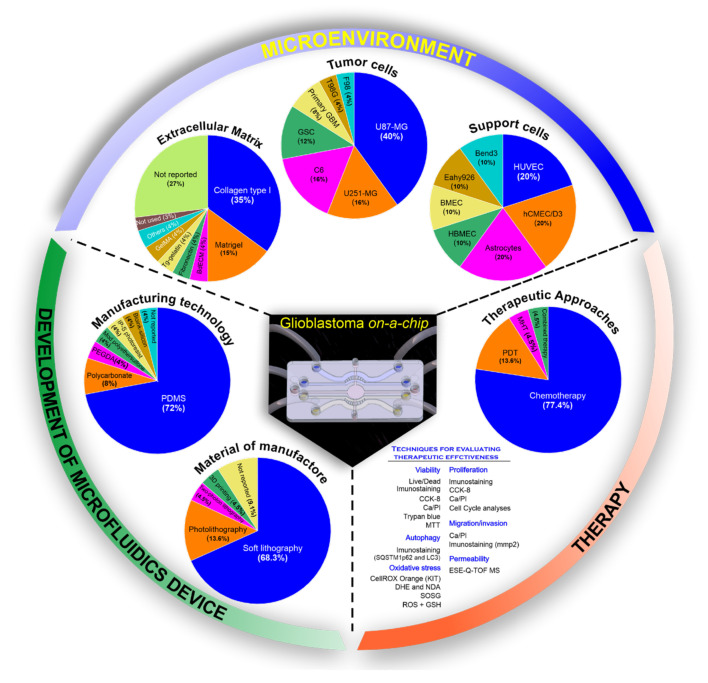
The systematic review identified three main points in glioblastoma on-a-chip for therapeutic application. The development of microfluidic devices was evaluated through the manufacturing technology and in the material used; the microenvironment, through the extracellular matrix, type of tumor cell used and support cells; and the therapy applied through different therapeutic approaches and their evaluation techniques in microfluidic devices.

**Table 1 cancers-14-00869-t001:** Microfluidic devices design e fabrication.

Study	Year	Manufacturing					Geometric Characteristics of Microdevices			
		Main Material of Device	Technology used	Mold Cast	Cover	Fabrication	Culture Region Shape	Device Dimensions	Device Structures	Complexity of Device
Li, Z.; et al. [39]	2021	PDMS	Soft lithography	SU-8 and ethyl lactate	No cover	In house	Semicircle	Top channel 0.5 × 2 × 11 mm^3^; Side channel 0.3 × 2 × 15 mm^3^; Center channel 100 × 900 × 11,000 µm^3^; Pore size 4 µm	Multi interfaces microdevice that consists in 3 layers: 1 channel at the top, and center, 2 channels bottom	++
Zhang, Q. et al. [40]	2020	PDMS	Soft lithography	SU-8	PDMS	In house	Circle	Width 60 μm; Height 100 μm; Length 150 μm	Inlet for flow injection and outlet for flow aspiration	+
Tricinci, O. et al. [41]	2020	IP-S photoresist	Two-photon lithography	SU-8	Glass	In house	Perimetric cylindrical pillars	Diameter 50 μm; Thickness 2 μm; Length 150–800 μm; Pores 5 μm	Arrangement of 10 microtubes, 2 flat ends, and a central cylindrical region with pores	+++
Samiei, E. et al. [42]	2020	PDMS	Soft lithography	SU-8	Coverslip	In house	Rectangular	Thickness 200 μm	4 parallel compartments (posts with gaps separate the adjacent compartments)	+
Mamani, J.B. et al. [43]	2020	PDMS	Soft lithography	SU-8	Glass	SynVivo Inc., Alabama, USA	Circle	Outer channel width 200 μm; depth (height) 100 μm; slit spacing 50 μm; travel (space between channels) of 50 μm	1 apical chamber; channels (2 external and 1 internal)	++
Yi, H. G. et al. [44]	2019	GBM-bioink; HUVEC-bioink; silicon	3D-printing system	NA	Glass	In house	Circle	NR	NA	+++
Qu, C. et al. [45]	2019	PDMS	NR	NR	Glass	In house	NR	NR	CGG unit and an open array of parallel chambers	++
Pang, L. et al. [46]	2019	PDMS	Soft lithography	SU-8 and silicon	PDMS	In house	Squares	Capture channel width 400 μm, height 25 μm; culture chamber width 2000 μm, height 25 μm, length 4500 μm; the microwell length 100 μm, width 100 μm; height 75 μm; pore 1 was 2 μm broader than pore 2	Channels (4 output and 1 input), pore and microchannels arrays	++
Burić, S. S. et al. [47]	2019	NR	NR	NR	NR	BEONCHIP, Zaragoza, Spain	Rectangular	NR	1 central and 2 lateral microchannels	+
Ma, J. et al. [48]	2018	PDMS	Soft lithography	SU-8	Glass	In house	Rectangular with circle array	CGG height/width 300 μm; open chambers width 5 mm, height 2 mm; pitch 4 mm	Open system CGG with parallel chambers in the form of a 4 × 4 array	++
Lin, C. et al. [49]	2018	PDMS and polycarbonate	Soft lithography	SU-8	PDMS	In house	Rectangular	Height microchannels 318.63 µm; pores 3 µm	3 parallel microchannels (2 outside and inside chambers)	+
Jo, Y. et al. [50]	2018	PDMS	Soft lithography	SU-8	Glass	In house	Rectangular	Length 2400 mm; width 4 mm	Sinuous microchannel with 7 folds; input and output channel	+
Akay, M. et al. [51]	2018	PEGDA hydrogel	Photolithography; light laser	NA	Glass	In house	Circle	Diameter of microwells 360 µm and microfluidic channels 100 µm, narrowing to 50 µm at the opening of microwells	2 inlets; 1 outlet; 7 microfluidic channels; 9–11 microwells per channel	++
Liu, H. et al. [52]	2017	PDMS	Soft lithography	SU-8	Glass	In house	Rectangular	Length 12 mm; width 1.2 mm; height 700 µm; diameter channel 600 µm	1 channel with 1 inlet and outlet chamber	+
Lee, J. M. et al. [53]	2017	PDMS	Soft lithography	SU-8	Glass	In house	Square	The thickness of the microchamber 250 μm and their bridge 40 μm	4 square-shaped microchambers and 8 bridge microchannels	++
Jie, M. et al. [54]	2017	PDMS and HF	Soft lithography	SU-8	PDMS	In house	Oval	NR	Serpentine porous hollow fibers embedded into a curved channel in the top layer, and 2 horizontally aligned oval chambers in the bottom layer of the chip with a connection array between chamber	+++
Zervantonakis, I. K. et al. [55]	2016	PDMS	Soft lithography	NR	Glass	In house	Rectangular	Channel width 2.5 mm and height 170 μm	8 pores interconnected with microchannels	+
Shao, X. et al. [56]	2016	PDMS and polycarbonate membrane	Photolithography	SU-8	NR	In house	Rectangular	μBBB channels-top layer: length 1 cm, width 2 mm, depth 100 µm; sub-layer: depth 100 µm; connection: depth 10 µm, width 1 mm; PM: thickness 10 µm and pore size of 0.4 µm; μSPE-one straight channel (22 mm length × 2 mm width × 80 μm depth); micropillar arrays (30 μm width intervals)	μBBB module: 2 PDMS sheets and a PM; μSPE module: 1 straight channel with micropillars arrays	+++
Gallego-Perez, D. et al. [57]	2016	PDMS	Photolithography	SU-8 and SPR950/SF6 nanowires	NR	In house	Rectangular	2 μm × 1 μm with 2 μm spacing	Arrays of parallel ridges	++
Xu, H. et al. [58]	2015	PDMS	Soft lithography	SU-8	Glass	In house	Rectangular	Upper/lower thickness layer: 190/100 μm	Microstructures with different heights	+
Yoon, H. et al. [59]	2014	PDMS	Soft lithography	SU-8	Glass	In house	Square	Reservoirs with diameter of inlet 2 mm and outlet 4 mm; channel 170 μm thick	Chambers (4 inlets and 4 outlets smaller); single test arena	++
Lou, X. et al. [60]	2014	PDMS	Soft lithography	SU-8	Glass	In house	Square	Microchannels height 33 µm; culture channels height 100 µm; filter layer, 3 different channel heights (15, 33, and 51 µm)	3 layers: glass (top), cell (middle) and filter (bottom)	+++

Abbreviations: PDMS: polydimethylsiloxane; IP-S: polymer photoresist; GBM-bioink: bioink of glioblastoma cells; GBM: glioblastoma multiform; HUVEC-bioink: bioink of HUVEC cells; HUVEC: human umbilical vein endothelial cells; NR: not reported; PEGDA: poly-(ethylene glycol) diacrylate (MW 700 Da); HF: modified polyethersulfone (mPES); SU-8: epoxy-based negative photoresist; NA: not applicable; SPR950/SF6 nanowires: nanowires SPR950 (~200 nm); BEONCHIP: biomimetic environment on chip (Spain); CGG: concentration gradient generator; μBBB: reconstruction BBB structure and 3D brain microenvironment; BBB: blood-brain barrier; PM: polycarbonate membrane; μSPE: solid-phase extraction on-chip.

**Table 2 cancers-14-00869-t002:** Characteristics of the cells used on a chip.

Study	Cell Type	Origin	Cells Bank or Primary Cells	Culture Media	Media Components	Complexity
Li, Z.; et al. [39]	BMEC	Rat	Primary cells	ECM	NR	++
	astrocytes			DMEM	10% FBS; 1% P/S	
	HepG2	Human	Cell Bank of the Chinese Academy of Sciences			
	U87-MG					
Zhang, Q. et al. [40]	U87-MG	Human	Cancer Institute & Hospital Chinese Academy of Medical Science, Beijing, China	MEM	10% FBS; 1% P/S; Earle’s salts; L-glutamine	+
Tricinci, O. et al. [41]	hCMEC/D3	Human	Merck Millipore (Massachusetts, MA, EUA)	EndoGRO-MV	EndoGRO-MV Supplement kit; 1% P/S	+++
	Primary astrocytes		Innoprot (Bizkaia, Spain)	DMEM high glucose	5% FBS; 3% astrocyte growth supplement; 1% L-glutamine; 1% sodium pyruvate; 1% P/S	
	U87-MG and U87-MG-GFP (spheroids)		ATCC Cellomix (Manassas, VA, USA)	DMEM high glucose	10% FBS; 1% L-glutamine; 1% sodium pyruvate; 1% P/S	
Samiei, E. et al. [42]	U87-MG/KD/SC	Human	ATCC (Manassas, VA, USA)	DMEM high glucose	10% FBS; 1% P/S	+
	U251-MG/KD/SC		Creative Bioarray-CSC-6321W			
Mamani, J.B. et al. [43]	C6	Rat	Cell Bank of Rio de Janeiro, Brazil	RPMI-1640	10% FBS; 1% P/S	+
Yi, H. G. et al. [44]	U87-MG	Human	ATCC (Manassas, VA, USA)	DMEM high glucose	10% FBS; 1% P/S; 1% L-glutamine;	++
	HUVEC		Promocell (Heidelberg, Germany)	ECM-2	NR	
	GBM isolated from patient		Primary cells	DMEM	10% FBS; 1% P/S;	
Qu, C. et al. [45]	U87-MG spheroids	Human	ATCC (Manassas, VA, USA)	DMEM (free red-phenol)	2,5% FBS	++
Pang, L. et al. [46]	U251-MG	Human	Chinese Academy of Sciences (Shanghai, China)	DMEM	10% FBS; 1% P/S	++
	Induced U251-MG			SFM/DMEM-F12	1% B-27; 20 ng/mL recombinant human EGF; 20 ng/mL FGF; 10 ng/mL LIF	
Burić, S. S. et al. [47]	TMZ and BCNU resistant C6 (RC6)	Rat	ATCC (Manassas, VA, USA)	DMEM	10% FBS; 2 mM L-glutamine; 4.5 g/L Glucose; 5000 U/mL penicillin; 5mg/mL streptomycin	++
Ma, J. et al. [48]	U87-MG	Human	Hui Chi Chen Biotechnology Co., Ltd., Shanghai, China	DMEM	10% FBS	+
Lin, C. et al. [49]	HBMEC	Human	Sciencell Corporation (Carlsbad, CA, USA)	ECM complete	5% FBS, 1% P/S, 1% ECCS	++
	Eahy926		ATCC (Manassas, VA, USA)	RPMI-1640	10% FBS, 1% PS	
	GSCs from GBM patients		The Second Affiliated Hospital of Soochow University	ECM complete	20 ng/mL EGF; 20 ng/mL FGFb; 2% B-27	
Jo, Y. et al. [50]	T98G	Human	Korean Cell Line Bank, Seoul, Korea	DMEM	10% FBS, 1% Penicillin	+
Akay, M. et al. [51]	Patient’s primary GBM	Human	UTHealth and Memorial Hermann, Texas Medical Center, Houston, TX, USA	Supplemented EBM (EGM-2)	FBS; hydrocortisone; GA-1000; VEGF; hEGF; hFGF-B; R^3^-IGF-1; acid ascorbic	+
Liu, H. et al. [52]	HUVEC	Human	Cancer Institute & Hospital Chinese Academy of Medical Science, Beijing, China	DMEM	10% FBS; 1% P/S	++
	U87-MG					
Lee, J. M. et al. [53]	MCF7	Human	NR	DMEM	10% FBS; 1%P/S	+
	U87-MG					
Jie, M. et al. [54]	Caco-2	Human	Cancer Institute and Hospital, Chinese Academy of Medical Sciences, Beijing, China	RPMI-1640	10% FBS;	++
	HepG2					
	U251-MG					
Zervantonakis, I. K. et al. [55]	F98-GFP	Rat	ATCC (Manassas, VA, USA)	RPMI-1640	10% FBS; 1% P/S	++
	Bend3	Mice				
Shao, X. et al. [56]	hCMEC/D3	Human	Institute COCHIN, Paris, France	RPMI-1640	10% FBS; 100 µg/mL P/S and 1.5 μM hydrocortisone	++
	U251-MG		NR	DMEM	10% FBS; 100 µg/mL P/S	
Gallego-Perez, D. et al. [57]	GSCs derived tumor: GBM157 and GBM528	Human	The Ohio State University	DMEM-F12	B27; 2.5 µg/mL heparin; 20 ng/mL FGFb and 20 ng/mL EGF	+
Xu, H. et al. [58]	U87-MG	Human	Cell Bank of the Chinese Academy of Sciences, Shanghai, China	DMEM	10% FBS	+
Yoon, H. et al. [59]	C6	Rat	ATCC (Manassas, VA, USA)	DMEM	10% FBS, 10,000 units/mL penicillin; 10,000 μg/mL streptomycin, and 25 μg/mL Fungizone	+
Lou, X. et al. [60]	C6	Rat	ATCC (Manassas, VA, USA)	DMEM	10% FBS; 1% P/S	+

Abbreviations: BMEC: brain microvascular endothelial cells primary; HepG2: liver hepatocellular carcinoma cell line; U87-MG: glioma cell line; hCMEC/D3: human cerebral microvascular endothelial cells; U87-MG-GFP: glioma cell line expressing green fluorescent protein; U251-MG: glioma cell line; U87/251-MG/KD/SC: Atg7 knockdown (KD) and scrambled (SC) U251 and U87 cells; C6: glial tumor of rat; HUVEC: human umbilical vein endothelial cells; GBM: glioblastoma multiform; TMZ: temozolomide; BCNU: bis-chloroethyl nitrosourea; RC6: TMZ and BCNU resistant C6; HBMEC: human brain microvascular endothelial cells; Eahy926: immortalized human vascular endothelial cells; GSCs: glioma stem cells; T98G: glioblastoma cell line; MCF7: breast cancer cell line; Caco-2: colorectal adenocarcinoma cell line; F98-GFP: glioblastoma cell line expressing green fluorescent protein; Bend3: mouse brain endothelial cell line; GBM157: cell clone isolated of patient; GBM528: cell clone isolated of patient; ATCC: American Type Culture Collection; NR: not reported; ECM: endothelial cell medium; DMEM: Dulbecco’s modified Eagle medium; MEM: minimum essential media; EndoGRO-MV: supplement kit containing 5% FBS, 5% l-glutamine, 0.2% EndoGRO-LS, 5 ng/mL–1 rhEGF, 1 µg/mL–1 hydrocortisone hemisuccinate, 0.75 U/mL–1 heparin sulfate, 50 µg/mL–1 ascorbic acid; RPMI-1640: Roswell Park Memorial Institute 1640 media culture; ECM-2: endothelial cell medium-2; SFM/DMEM-F12: Dulbecco’s modified Eagle medium-F12 containing neural stem cell medium serum-free; ECM complete: endothelial cell medium complete; EBM: endothelial basal medium; EGM-2: endothelial cell growth medium-2; FBS: fetal bovine serum; P/S: penicillin and streptomycin; EGF: epidermal growth factor; FGF: fibroblast growth factor; LIF: leukemia inhibitory factor; ECCS: endothelial cell growth supplement; ECC: endothelial cell growth; FGFb: basic fibroblast growth factor; B-27: supplement of medium; GA-100: gentamicin sulfate-amphotericin; VEGF: vascular endothelial growth factor; hEGF: human epidermal growth factor; hFGF-β: human fibroblast growth factor-basic recombinant; R^3^-IGF-1: long arginine 3-IGF-1.

**Table 3 cancers-14-00869-t003:** 3D culture development of glioblastoma model in microfluidic devices.

Study	Extracellular Matrix			Cells Type (Cells/mL)	Culture Time (d)	Cultivation Method on the Device	Medium Change (h)/Flow (μL/min)	Complexity
	Type of Matrix	Concentration (mg/mL)	Volume (µL)					
Li, Z.; et al. [39]	COL1	6	NR	Astrocytes (5 × 10^5^); HBMEC (1 × 10^5^); HepG2 (1 × 10^6^); U87 (NR)	2.5	COL1 was perfused into the channels (10 min), following by seeded astrocytes. After 12 h, BMECS were seeded in the same channels, 24 h later, HepG2 cells were perfused in the upper chamber and more 24 h, U87-MG cells were introduced into the lower right channel	NR	+++
Zhang, Q. et al. [40]	NA	NA	NA	U87 (1 × 10^4^ cells/cm^2^)	0.125, 0.25, 6, 0.5, 0.75	The adherent target single cell in trypsin region was digested, and the extraction process was recorded by microscope camera.	Injection:10 μL/min; aspiration: 40 μL/min	+
Tricinci, O. et al. [41]	NA	NA	NA	hCMEC/D3 (3 × 10^4^ cells/cm^2^); primary astrocytes (1 × 10^4^/cells cm^2^); U87 (2 × 10^4^ cells/30 µL)	5	hCMEC/D3 cells were seeded inside microtubes. After 5 days, the human primary astrocytes were seeded on the outside part of the tubes, and U87-MG cells were seeded in the MRCSs, after 5 days of cell growth	NR/4.7 × 10^3^	+++
Samiei, E. et al. [42]	COL1	3	NR	U251 and U87 (10^6^ viability)	4	COL1/cell suspension was injected into the channel for 45 min (invasion study) or overnight (viability study), and the treatment was started the day after.	NR	+
		4		U87 (5 × 10^6^ invasion)				
Mamani, J.B. et al. [43]	Matrigel	9-12	15	C6 (10^7^)	2	Matrigel was injected into the central channel for 2 h. Then, C6 cells were injected into the external channel	4/5	+
Yi, H. G. et al. [44]	BdECM	10	NR	U87, GBM from patients and HUVEC (5 × 10^6^)	7	The cell-laden bioinks were encapsulated with GBM cells or HUVECs into pre-gel solutions of BdECM or collagen.	24/NR	+++
	COL1							
Qu, C. et al. [45]	COL1	1.5	NR	U87 (3.5 × 10^5^)	2	U87 cells were seeded in the channel. Then, arrays were generated on the inverted PDMS surface, using COL1 was used as the U87 spheroid encapsulating ECM.	24/0.5	++
Pang, L. et al. [46]	NA	NA	NA	U251 (0.25–2.5 × 10^4^)	10	Pluronic F127 was injected from the inlet into the chambers for 2 h at 20 °C. Cells were seeded into the chambers for the 20 s from the inlet (20 μL/min), using different driving infusion flow rates (25–150 μL/min) to separate the single-cells, cultured at a slow perfusion rate (5 μL/min)	Half of the medium 24/5	+
Burić, S. S. et al. [47]	COL1	NR	10	RC6 (5 × 10^6^)	3	10 μL of the mixture of RC6 cells with COL1 was injected into the central chamber. After COL1 polymerization for 15 min, lateral microchannels were perfused with medium	2/NR	+
Ma, J. et al. [48]	COL1	1, 5	8	U87 (6 × 10^5^)	3	U87 cells spheroids were formed for 3 days in chamber lower, in following added COL1 for 45 min and PDMS surface was inverted to solidify	NA/0.5	++
Lin, C. et al. [49]	Fibronectin	NR	NR	HBMEC, Eahy926 and GSC (1.26 × 10^6^)	3	HBMECs and Eahy926 cells were seeded in the upper microchannels, the following day, the GSCs were cultured in the lower channels. Fibronectin was placed (12 h) for endothelial cell culture.	12/NR	++
Jo, Y. et al. [50]	Matrigel	0.1	NR	T98G (10^5^)	4	The microchannel was coated with a PDL or Matrigel solution for 3 h, and then the T98G cells were seeded. After cultivate, the flow was stopped for 24 h. Then shear stress of (0.1 dyn/cm^2^) was applied	72/NR	++
	PDL	0.1	NR					
Akay, M. et al. [51]	NA	NA	NA	GBM of patients (5 × 10^5^)	7	The GBM cells were seeded into both inlet channels simultaneously. The cells were captured in the microwells and cultured for 7 days	A half medium 48–72/NR	+
Liu, H. et al. [52]	TG-gelatin	NR	NR	U87 and HUVEC (10^7^)	3	TG-gelatin suspension was used for U87 cells culture. Then, these cells were injected into the channels with the PU fiber. After the gel polymerized and solidified for 40 min, the PU fiber was pulled out from the channels. Following, HUVEC cells were seeded into the lumen (4 h), and the chip was connected to the peristaltic pump.	NR/(0,5,10,20)	+++
Lee, J. M. et al. [53]	GelMA	10	20	MCF7 and U87 (2 × 10^6^)	5	MCF7 and U87MG cells were cultured in square-shaped microchambers	24/NR	++
Jie, M. et al. [54]	Matrigel	3.86	NR	U251, HepG2 and Caco-2 (10^6^)	NR	HF was coated with a Matrigel per 1 h at 4 °C. Then, 10 μL of the Caco-2 cells were seeded into the lumen. HepG2 and U251 cells were injected into chambers b and c (bottom layer) from the respective inlets. The inlet and outlet of the HepG2 cell chamber were stoppered. The inlet of the U251 cells chamber (b and c) was stoppered and the outlet was connected to a waste reservoir. After 24 h, the outlet of HF was stoppered and the inlet was connected to an infusion pump that continuously infused the medium (5 μL/h).	12/0.083	+++
Zervantonakis, I. K. et al. [55]	COL1	2	NR	F98-GFP (3 × 10^5^) and Bend3 (2 × 10^6^)	2	F98G cells were seeded in the top layer and Bend3 cells in the bottom layer, in the device containing COL1 for 48 h	NR/NR	++
Shao, X. et al. [56]	Matrigel	0.1	NR	hCMEC/D3 (5 × 10^6^)	3	μBBB model: hCMEC/D3 cells were seeded on the upper side of the membrane (to form cell monolayer) and inferior chambers the U251 cells were injected and encapsulated in agarose solution for 24 h.	24/15	+++
	Agarose	NR	NR	U251 (5 × 10^6^)			24/NA	
Gallegos-Perez, D. et al. [57]	NR	NR	NR	GSCs derived tumor: GBM157 and GBM528	NR	GSC, GBM157, and GBM528 clones neurospheres were dissociated and seeded on the microtextured chip surface (16 h of monitorization)	NR/NR	+
Xu, H. et al. [58]	COL1	NR	NR	U87 (5 × 10^4^ cell/cm^2^)	1	Cells were seeded into the center channel. The chip was then turned on its side for 5 min. Each chip was then incubated for either 24 h, 21% O_2_ (normoxic condition) or 0.2% O_2_, 94% N_2_ (hypoxic conditions). Cells were allowed to invade for 24 h.	NR/NR	+
Yoon, H. et al. [59]	COL1	NR	NR	C6	1	The chip was coated, using a solution of 0.01% COL1, then it was seeded with C6 cells for about 24 h.	NR/NR	+
Lou, X. et al. [60]	NA	NA	NA	C6 (2 × 10^6^)	Overnight	In the gas layer of CGG containing the C6 cells was introduced compressed air and nitrogen, generating an oxygen gradient from 1.3% (hypoxia range) to 19.1% (ambient air range).	NA	+

Abbreviations: COL1: chilled liquid type I collagen; NA: not applicable; BdECM: brain decellularized ECM; PDL: poly-D-lysine hydrobromide; TG-gelatin: gelatin transglutaminase; GelMA: gelatin methacrylate hydrogels; NR: not reported; HBMEC: human brain microvascular endothelial cells; HepG2: liver hepatocellular carcinoma cell line; U87: glioma cell line; hCMEC/D3: human cerebral microvascular endothelial cells; U251: glioma cell line; C6: glial tumor of rat; GBM: glioblastoma multiform; HUVEC: human umbilical vein endothelial cells; RC6: C6 resistant to TMZ and BCNU; Eahy926: immortalized human vascular endothelial cells; GSC: glioma stem cells; T98G: glioblastoma cell line; MCF7: breast cancer cell line; Caco-2: colorectal adenocarcinoma cell line; F98-GFP: glioblastoma cell line expressing green fluorescent protein; Bend3: mouse brain endothelial cell line; GBM157: cell clone isolated of patient; GBM528: cell clone isolated of patient; BMECS: brain microvascular endothelial cells; MRCSs: magnetically-responsive cage-like scaffolds; RPMI: Roswell Park Memorial Institute; PDMS: polydimethylsiloxane; ECM: endothelial cell medium; PU: polyurethane; HF: hollow fiber; μBBB: reconstruction BBB structure and 3D brain microenvironment; BBB: brain blood barrier.

**Table 4 cancers-14-00869-t004:** Glioblastoma therapeutic approach in site of microfluidic device.

Study	Therapeutic Approaches	Therapeutic Dose (µM)	Time of Treatment (h)	Evaluation Efficacy Treatment	Outcomes	Complexity
Li, Z.; et al. [39]	PTX	2.3 × 10^−3^	48	Live/dead; CCK-8 kit; mass spectrometry;	In the liver-brain system, the liver had enhanced cytotoxicity of CAP on U87 cells by 30% while having no significant effect on TMZ. However, the BBB system showed a 20% decrease in PTX cytotoxicity, already no significant effect was found on TMZ and CAP	++
	CAP	80				
	TMZ	40				
Zhang, Q. et al. [40]	5-fluorouracil	38.4	3, 6, 12, or 18	Calcein-AM/PI	All drugs no influence under proliferation and viability into 6 h, already the TMZ showed a reduction cell-matrix adhesion and their effect was less significant with the increase of the lactic acid.	++
	Actinomycin D	10				
	Allicin	200				
	TMZ without LA	0; 100; 300; 500				
	TMZ with LA	500 (TMZ); 0; 10^4^; 2 × 10^4^ (AL)	6			
Tricinci, O. et al. [41]	Ab-Nut-NLCs	400 µg/mL + EMF of 1.31 T	24	Live/dead; immunostaining (Ki-67);	Ab-Nut-NLCs capacity to cross the BBB and efficacy about 70% on the treatment	+++
Samiei, E. et al. [42]	TMZ and simvastatin	0; 100; 250; 500 (TMZ) and 0; 1; 5; 10 (simvastatin)	72	Live/dead; immunostaining (cleaved-caspase-3 and PARP, SQSTM1p62 and LC3);	The viability and invaded cells had a dose-dependent effect; U251 cells were more sensitive to the treatments than the U87 cells, showing more effectively TMZ (500 µM) than simvastatin (10 µM).	++
Mamani, J.B. et al. [43]	MHT and magnetic nanoparticles	10 mgFe/mL (20 µL) + 300 Gauss/305 kHz	0.16; 0.5	Live/dead	After MHT, the cell viability reduced by 20% and 100% after 10 and 30 min, respectively	+++
Yi, H. G. et al. [44]	CCRT combined with TMZ, CIS, KU, O^6^BG and MX	Different drug combinations: 950; TMZ; 950 CIS; 250 KU; 210 O^6^BG; 150 MX + 15 Gy	24 (1 h gamma irradiation)	CCK-8	The drug combination (TMZ, CIS, KU, O6BG, and MX) was more effective on the GBM-28-on-a-chip. However, the GBM-37-on-a-chip showed the highest resistance to the tested drugs.	+++
Qu, C. et al. [45]	TAM	10; 20; 30	24	Cell cycle analysis (acridine orange); calcein-AM/PI;	ER- α36 knockdown increased sensitivity of glioblastoma U87 cells to TAM and decreased autophagy in these cells. However, ER- α36 overexpression decreased TAM sensitivity and induced autophagy.	+
Pang, L. et al. [46]	Vincristine	1.25; 2.5; 5; 10; 20; 40; 80	0, 6, 12, 18, 24	Immunostaining (JC-1; Caspase-3)	Drug resistance in the induced of U251 spheres was higher than standard U251 cells and dose-dependent.	+
Burić, S. S. et al. [47]	TMZ	250	72	Calcein-AM/PI; ROS (CellROX Orange, Thermofisher, Massachusetts, MA, USA)	CoQ10 can suppress invasiveness, the epithelial to mesenchymal transition in RC6 cells, as also decrease ROS and when combined with TMZ, exerted a synergistic antiproliferative effect and is more cytotoxic than TMZ monotherapy.	++
	CoQ10	10				
Ma, J. et al. [48]	Resveratrol	0; 100; 200; 300	24; 48; 72	CCK-8; calcein-AM/PI; Immunostaining (Ki-67; vimentin and MMP2);	GBM responses in resveratrol + TMZ groups were better than single drug groups, showed enhanced inhibitor effects, in total invasive, and mesenchymal phenotype transition degree of GBM.	++
	TMZ					
Lin, C. et al. [49]	TMZ	0; 200; 400; 600; 800; 1.200	72	Live/dead; microchip electrophoresis and high-resolution melting	TMZ led to a 50% death rate of GSCs, as well as SU3-GSCs were more sensitive than U251-GSCs. The co-culture of the GSCs with the endothelial cells led to the GSCs chemoresistance against the TMZ	+
Jo, Y. et al. [50]	DOX	1μg/mL	72	Calcein-AM/PI	In the Matrigel-coated chip, tumor cell growth increased slowly, showing a chemoresistance in the DOX presence.	+
Akay, M. et al. [51]	TMZ	600	168	Trypan blue	The drug response was different between each patient’s cells, and drug combined (TMZ + BEV) resulted in a higher cell death than monotherapy, in which the TMZ was better efficient.	++
	BEV	7.5				
Liu, H. et al. [52]	Catechins	0; 250; 500; 750; 1000	48	ROS and GSH	Drugs displayed higher efficacy to U87 cells than HUVEC cells. The decrease of ROS and increase of GSH in cells were accelerated with the increase of antioxidants (mainly for α-lipoic acid), controlling the intracellular ROS level within its safety limit, and cell invasion was inhibited	++
	α-lipoic acid					
	Ascorbic acid					
Lee, J. M. et al. [53]	NIR laser irradiation gold nanorod	20 v/v% + 3A/4.27W	0.25	CCK-8; live/dead	Regardless of the cancer cell type (MCF7 and U87), viability was less than 10% after irradiation with NIR laser.	+++
Jie, M. et al. [54]	Combined CPT-11, TMZ and CP	6.25; 12.5; 25; 50 and 75 μg/mL	12	CCK-8; ROS and GSH by DHE and NDA; immunostaining (JC-1); live/dead; LC-MS flow cytometry;	Combined drugs (CPT-11 + TMZ shows the best results) showed growth inhibition effects and decrease cell viability.	++
Zervantonakis, I. K. et al. [55]	FUS and DOX-TS-liposomes	0.03; 0.1; 0.3; 1; 3 and 10 + 3.525 MHz/2.2 W	1	Immunostaining (DAPI; γ-H2AX; GFP)	DNA damage and tumor cell death were confined to the area of drug release, ~40.9 ± at the center and decayed to a baseline value of ~18.8% toward the edges of the cell chamber	+++
Shao, X. et al. [56]	Sunitinib	10	0, 24, 48	ESI-Q-TOF MS; live/dead; MTT	The drug permeability across BBB and their efficiency were better through the hCMEC/D3 monolayer	+
Gallego-Perez, D. et al. [57]	TMZ	0; 0.005; 0.050; 0.500 and 5	24; 48; 96	Live/dead	The cell viability decreased by 60% by TMZ (96 h) and 80% by anti-miR363, this drug also affects cell motility in the first 48h. So, TMZ + anti-miR363 combined decreased viability by 80–90%.	++
	Anti-miR363	2 and 5 NEP				
Xu, H. et al. [58]	Normoxic and inhibited by siRNA HIF1α and HIF2α	21%	24 or 48	Immunostaining (Ki-67; MMP2; Zeb1/2; Snail/Slug; Twist; HIF1/2α; vimentin); RT-qPCR-RT (GLUT1, VEGFA, EDN1; EPO; MMP2 and MMP9); Western blotting (Twist; MMP2; MMP9);	Hypoxia activates mesenchymal transition and enhances cell motility in GBM in a HIF-dependent manner, and this process can be attenuated by pharmacological blockade of HIFα. Antiangiogenic therapy associated with HIFs inhibitors can delay tumor progression	++
	Hypoxic and inhibited by siRNA HIF1α and HIF2α**	0.2 and 1% (O_2_)				
Yoon, H. et al. [59]	PDT by MB-PEGDMA PAA NPs	MB–PEGDMA PAA NPs, with MB (2.1; 5.5; 12.1 µmol/g) + (~625 nm/35.2 mW; LED light doses 0 to 39.2 J/cm^2^)	0-0.35	Live/dead; singlet oxygen sensor green (ROS)	C6 cells killing effects of the various MB–PEGDMA PAA NPs were light-dose-dependent	+++
Lou, X. et al. [60]	PDT by MB combined with hypoxic conditions	0–10 (MB); 0–21% (O_2_) + (637 nm; 0–9.5 mW; light dose 42.8 J/cm^2^)	0.5	Live/dead	Cell viability decreased to around 0% with the increase of light power until 9.5 mW. Samples with higher drug concentrations had a viability drop than a lower concentration.	+++

Abbreviations: PTX: paclitaxel; CAP: capecitabine; TMZ: temozolomide; LA: lactic acid; Ab-Nut-NLCs: antibody-functionalized nanostructured lipid carriers loaded with nutlin-3a; MHT: therapy of magnetic hyperthermia; CCRT: concurrent chemoradiation; CIS: cisplatin; KU: improved ATM kinase-specific inhibitor; O^6^BG: O^6^-benzylguanine; MX: methoxyamine; TAM: Tamoxifen; CoQ10: coenzyme Q10; DOX: doxorubicin; BEV: bevacizumab; NIR: near-infrared; CPT-11: irinotecan; CP: cisplatin; FUS: focused ultrasound; DOX-TS-liposomes: doxorubicin encapsulated temperature-sensitive liposome formulation; siRNA: small interfering RNA; HIF1-α/HIF2-α: hypoxia-inducible factor 1α/2α; PDT: photodynamic therapy; MB–PEGDMA PAA NPs: MB conjugated polyacrylamide nanoparticles (PAA NPs), with a polyethylene glycol dimethacrylate (PEGDMA, Mn 550) cross-linker; MB: methylene blue; EMF: external magnetic field; NEP: nanochannel-based electroporation; LED: light-emitting diode; CCK-8: cell counting kit-8; Calcein-AM: calcein acetoxymethyl ester; PI: propidium iodide; PARP: poly-ADP ribose polymerase; SQSTM1 p62: sequestosome 1 gene; LC3: microtubule-associated proteins 1A/1B light chain 3B (hereafter referred to as LC3); JC-1: 5,5′,6,6′- tetrachloro-1,1′,3,3′-tetramethyl benzimidazole-carbocyanine iodide;; ROS: reactive oxygen species; MMP2: matrix metalloproteinase-2; GSH: glutathione; DHE: dihydroethidium; NDA: 2,3-naphthalenedicarboxaldehyde;; LC-MS: liquid chromatography–mass spectrometry; DAPI: 4′,6-diamidino-2-phenylindole, dihydrochloride; GFP: green fluorescent protein; ESI-Q-TOF MS: electrospray ionization quadrupole time-of-flight mass spectrometer; MTT: 3-(4,5-dimethylthiazol-2-yl; RT-qPCR: quantitative reverse transcription PCR; BBB: blood-brain barrier; GBM-28/37: patient GBM derived cell strains 28 and 37; ER-α36: estrogen receptor alpha-36; RC6: C6 resistant to TMZ and BCNU; SU3-GSCs: GSCs derived from SU3 of cell line; U251-GSCs: GSCs derived from U251 cell line; GSC: glioma stem cells; HUVEC: human umbilical vein endothelial cells; MCF7: human breast carcinoma cells; hCMED/D3: human cerebral microvascular endothelial cells;. Note: ** pharmacologic inhibition of HIFs was achieved using an inhibitor of HIF1α-mediated transcription (methyl-3-[[2-[4-(2-adamantyl)phenoxy]acetyl]amino]-4-hydroxybenzoate) (Santa Cruz Biotechnology, Santa Cruz, CA, USA) or HIF2α translation (methyl-3-(2-(cyano(methylsulfonyl)methylene)hydrazino)thiophene-2-carboxylate) (Merck Millipore, Darmstadt, Germany) at a concentration of 30 μM in DMSO.

## Data Availability

Not applicable.

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
