# Peer review of "The Advances in Glioblastoma On-a-Chip for Therapy Approaches"

_cancers, 2022, doi:10.3390/cancers14040869_

Round 1

Reviewer 1 Report

Review

The systematic review titled “The advances in Glioblastoma-on-a-Chip for therapy approaches” by Arielly H. Alves and co-authors, concentrates on microfluidic devices for the reconstitution of the glioblastoma microenvironment in a 3D model, aiming at therapeutic approaches.

The authors included publications of the last 10 years, considering the period between September 2011 and September 2021, indexed in PubMed, and Scopus with a total of 22 filtered/screened unduplicated full-text articles (studies from USA and China), that were discussed deeply in this review.

This review is an exhaustive, clear, and informative review that is also well organized.

Minor

There are little typo errors:

For example in the introduction: “is one of the most fatal malignnat diseases in humans.”

Or in line 102 there is a full stop between the sentence and the References

“the cellular microenvironment.[21-23].”

Finally, Fig 1 could be changed including only the last panel (included studies), while I would enlarge Fig 2 using the whole page.

Author Response

Reviewer #1

The systematic review titled “The advances in Glioblastoma-on-a-Chip for therapy approaches” by Arielly H. Alves and co-authors, concentrates on microfluidic devices for the reconstitution of the glioblastoma microenvironment in a 3D model, aiming at therapeutic approaches.

The authors included publications of the last 10 years, considering the period between September 2011 and September 2021, indexed in PubMed, and Scopus with a total of 22 filtered/screened unduplicated full-text articles (studies from USA and China), that were discussed deeply in this review.

This review is an exhaustive, clear, and informative review that is also well organized.

Minor

  1. There are little typo errors:

For example in the introduction: “is one of the most fatal malignnat diseases in humans.”

Or in line 102 there is a full stop between the sentence and the References

“the cellular microenvironment.[21-23].”

Answer:  Thank you for your time and dedication in reviewing the manuscript. Both errors were fixed, and the manuscript was checked again.

  1. Finally, Fig 1 could be changed including only the last panel (included studies), while I would enlarge Fig 2 using the whole page.

Answer:  Thank you for your suggestion. We enlarged Fig 2, using the whole page, but we will maintain Fig 1 with the flowchart that represents the all-screening process of the systematic review study according to the PRISMA guideline.   

Reviewer 2 Report

The authors present an interesting and comprehensive review of a highly relevant topic in glioblastoma research.

The figures are of outstanding quality.

References are cross-checked and complete.

The only minor remark touches a comprehensive paragraph on the limits of this manuscript and the analysis as performed by the authors that should be included at the end of the discussion.

Apart from that, the authors should be applauded for their excellent work.

Author Response

Reviewer #2

The authors present an interesting and comprehensive review of a highly relevant topic in glioblastoma research.

The figures are of outstanding quality.

References are cross-checked and complete.

  1. The only minor remark touches a comprehensive paragraph on the limits of this manuscript and the analysis as performed by the authors that should be included at the end of the discussion.

Apart from that, the authors should be applauded for their excellent work

Answer: Thank you for your suggestion. We added one paragraph at the end of the discussion section of the manuscript about the limitations of the study.
